# Reactive Oxygen and Sulfur Species: Partners in Crime

**Neil W. Blackstone**

Department of Biological Sciences, Northern Illinois University, DeKalb, IL 60115, USA; neilb@niu.edu

**Abstract:** The emergence of complexity requires cooperation, yet selection typically favors defectors that do not cooperate. Such evolutionary conflict can be alleviated by a variety of mechanisms, allowing complexity to emerge. Chemiosmosis is one such mechanism. In syntrophic relationships, the chemiosmotic partner benefits simply from exporting products. Failure to do this can result in highly reduced electron carriers and detrimental amounts of reactive oxygen species. Nevertheless, the role of this mechanism in the history of life (e.g., the origin of eukaryotes from prokaryotes) seems questionable because of much lower atmospheric levels of oxygen and a largely anaerobic ocean. In this context, the role of sulfur should be considered. The last eukaryotic common ancestor (LECA) was a facultative aerobe. Under anaerobic conditions, LECA likely carried out various forms of anaerobic metabolism. For instance, malate dismutation, in which malate is both oxidized and reduced, allows re-oxidizing NADH. The terminal electron acceptor, fumarate, forms succinate when reduced. When oxygen is present, an excess of succinate can lead to reverse electron flow, forming high levels of reactive oxygen species. Under anaerobic conditions, reactive sulfur species may have formed. Eliminating end products may thus have had a selective advantage even under the low atmospheric oxygen levels of the Proterozoic eon.

**Keywords:** atmospheric oxygen; chemiosmosis; eukaryotes; history of life; mitochondria; syntrophy

## 1. Introduction

Cooperation is essential to the emergence of evolutionary complexity. Indeed, much of the history of life can be understood in terms of this basic framework [1–3]. Throughout the history of life, biological units (e.g., genes, prokaryotes, eukaryotes) have banded together to form higher level units (e.g., chromosomes, eukaryotic cells, and multicellular organisms). While major and minor transitions have occurred repeatedly, these were not necessarily straightforward because evolutionary conflicts hamper cooperation. For instance, as prokaryotes banded together to form eukaryotic cells, selection at the higher level favored cooperation, but selection at the lower level favored defection, i.e., abandoning the common good to engage in selfish replication. Of course, these evolutionary dynamics are not limited to the history of life but occur in modern symbioses and multicellular organisms as well.

For cooperation and complexity to emerge, evolutionary conflicts must be arbitrated or mediated. Conflict mediation increases the variance at the higher level, allowing greater scope for selection at this level to favor cooperative groups, or decreases the variance at the lower level, limiting the origin of defectors [4,5]. Conflict arbitration acts similarly but occurs very early in an evolutionary transition prior to the formation of higher-level units, when only lower-level selection is acting [6,7]. For the most part, the focus here will be on the latter: the incipient steps of a transition when only lower-level units were present. Signaling among the lower-level units may be important in this context [7].

In particular, metabolic or "redox" signaling related to chemiosmosis may have had a major impact on the history of life. Consider the banding together of prokaryotes to form eukaryotes. While there are numerous models for this process [8–11], most models converge on several salient points. First, the initial association of lower-level units was

based on some sort of syntrophy, i.e., "feeding together", where the waste of one partner was the substrate for the other. Second, the partner that became the mitochondrion had a functional electron transport chain that it used to carry out chemiosmosis.

Chemiosmosis refers to a form of energy conversion that is found in virtually all living things [12]. In a cell or organelle surrounded by a membrane that is impermeable to charged ions, metal-containing protein complexes (i.e., electron carriers) pass electrons from one to another and in the process extrude protons. An excess of protons thus surrounds the membrane of the chemiosmotic cell or organelle. When these protons return through the membrane via ATP synthases, they impart energy to a tiny mechanical mechanism, joining adenosine diphosphate (ADP) and inorganic phosphate to form adenosine triphosphate (ATP).

In any syntrophic association, the consumption of excreted metabolites by a partner provides a benefit to the other partner simply because lowering the concentration of these metabolites alleviates end-product inhibition. Any chemical reaction will be slowed or even stopped by a build-up of products. Nevertheless, the consequences of end-product inhibition differ depending on the nature of the metabolic reaction. Substrate-level phosphorylation will be slowed by an accumulation of ATP, but detrimental by-products will not necessarily be formed. On the other hand, given that a fundamental aspect of chemiosmosis is the separation of hydrogen atoms into electrons and protons, a buildup of end-products (e.g., ATP) will have more serious consequences, for example, slowing the progress of electrons through the electron transport chain so that these electrons back up on the electron carriers. Highly reduced electron carriers typically cast-off some of these electrons. Under aerobic conditions, molecular oxygen will accept these electrons and form reactive oxygen species (i.e., partially reduced forms of oxygen, ROS), which can be detrimental in large quantities. Thus, in a syntrophic relationship, the chemiosmotic partner reaps a large benefit simply by getting rid of excess product. Crucially, the potential cost to the chemiosmotic partner of retaining this product exceeds whatever benefit it provides. Such a calculus can drive cooperation in cases in which it would not otherwise occur [13–15].

However, as nicely summarized by Martin et al. [16], atmospheric oxygen remained quite low (perhaps 1% of present atmospheric levels) for the entire Proterozoic eon, during which eukaryotes originated and then repeatedly evolved multicellularity. Further, for much of this time the oceans, in which most organismal evolution occurred, were not only anaerobic but euxinic, i.e., rich in $H_2S$. Under these conditions, oxidative phosphorylation may have been used only intermittently and was thus less likely to drive a syntrophic relationship toward an endosymbiosis. The implications of these geological data will be further examined herein. Aerobic and anaerobic hypotheses for the origin of eukaryotes will be explored as well as the possible contributions of chemiosmosis. The goal of this discussion is not to develop an explicit model of the symbiosis but rather to explore the possible avenues of conflict mediation in the Proterozoic. Finally, the key question will be addressed: can chemiosmosis mediate evolutionary conflict even under anaerobic conditions?

## 2. The Evolution of Eukaryotes

Using features of modern eukaryotes, phylogenetic methods can be used to reconstruct the character states of the last eukaryotic common ancestor (LECA) [17]. Nevertheless, these data cannot provide insight into events that preceded LECA, e.g., what were the character states of the first eukaryotic common ancestor (FECA) and what was the path from FECA to LECA? Various models of eukaryogenesis [8–11] may be useful in this regard. Further, the recently discovered "Asgard" archaea are of considerable interest as models of host characteristics [18,19]. In this regard, it is sometimes overlooked that these are crown-group archaea and thus separated from putative hosts by perhaps 2 billion years of evolution. Generally, modern eukaryotic physiology and particularly bioenergetics may provide the best guide to illuminating the characteristics of the first eukaryotes and the path from FECA to LECA.

While eukaryotes are diverse in terms of genes, genomes, life history, and structure, they are highly stereotypical in terms of bioenergetics [16,20]. In large part, eukaryotic bioenergetics derive from endosymbiotic mitochondria, which are a shared derived character of eukaryotes [17]. While some eukaryotes have secondarily lost their mitochondria, to my knowledge none of these has subsequently re-acquired them, e.g., by taking up another mitochondrion-containing eukaryote. On the other hand, there are eukaryotes that primitively lack plastids, and some of these have taken up plastid-containing eukaryotes. Further, modern eukaryotic symbioses (e.g., corals, lichens, mycorrhizal fungi) typically involve oxygenic photosynthesis rather than oxidative phosphorylation. Thus, there are no recent endosymbiotic events that convincingly parallel the mitochondrial symbiosis. Nevertheless, some understanding of ancient events in eukaryotic history can be gained by cautiously examining putative vestiges of these processes that are found in the physiology of modern eukaryotes. While some of the details of mitochondrial physiology were elucidated prior to the chemiosmotic theory (e.g., [21]), chemiosmosis provides a powerful rationalization for these details [12].

Finally, in recent decades there has been increasing appreciation of the diversity of organelles of mitochondrial origin, e.g., mitosomes, hydrogenosomes, and anaerobic mitochondria, which to some extent exhibit divergent physiologies [16]. Additionally, the path of evolution from LECA to modern eukaryotes has become clearer. As described by Martin et al. [16], this path reflects specialization followed by loss of unneeded capabilities: "Mitochondria, hydrogenosomes, mitosomes, anaerobic mitochondria, and their various intermediate forms are variants that arose from a common ancestral organelle by gene loss and specialization".

## 3. From FECA to LECA

What then of the path from FECA to LECA? It may be impossible to precisely discern this pathway, but several considerations can constrain the possibilities. The path from LECA to modern eukaryotes was one of specialization and narrowing of function. How does this proceed? When an organism ceases to use a function, there is no fitness penalty for the loss of genes that influence the underlying functional capacity. Unless these genes are constrained by pleiotropy, they can thus undergo mutational decay without any purifying selection at the organismal level. While some instances of re-derived complex adaptations have been found [22], something as complex as the electron transport chain, which only evolved once in the history of life, would seem extremely unlikely to be regained after being lost.

In all likelihood, the protomitochondria that preceded LECA were therefore capable of the full range of bioenergetics found in modern mitochondria, including aerobic metabolism (e.g., Krebs cycle, chemiosmosis) and the combined anaerobic capabilities as well [16]. All the mitochondria on the pathway from FECA to LECA must have retained these capabilities. While any particular function may have only been used intermittently, there could not have been long periods of time where some functions were not used at all. In this context, consider the most widely known model for the first steps in eukaryogenesis, the "hydrogen hypothesis" [23]. Briefly, in an anaerobic environment, heterotrophic, sometimes aerobic, bacterial protomitochondria carried out glycolysis. As with anaerobic metabolism in general, re-oxidizing NADH was necessary. Here, the protomitochondria did this by converting pyruvate to $CO_2$, $H_2O$, and $H_2$. Essentially, the electrons from NADH were funneled onto protons to make $H_2$ as well as perhaps fatty acids. These protomitochondria formed a syntrophic relationship with autotrophic archaea, which used these waste products so that $H_2$ was now the electron donor and $CO_2$ was reduced to both sugars and methane. The latter was released as waste. In time, the bacterial community became endosymbiotic. While it may seem that the bacteria were initially useful to the archaea but not the converse, i.e., that this was a commensal relationship, when end-product inhibition is considered, the initial advantages to the bacteria become clear, and properly this should be regarded as a mutualistic symbiosis.

Nevertheless, if FECA metabolized only in this way for many generations, other metabolic capabilities of protomitochondria would have been lost via mutation unless there was significant pleiotropy. One may thus surmise that other aspects of this syntrophic relationship developed in parallel, perhaps stemming from day/night alterations in environmental conditions. In the presence of cyanobacterial photosynthesis, for example, molecular oxygen may have been available, and the protomitochondria may have metabolized acetyl CoA via the Krebs cycle and used the electron transport chain to re-oxidize NADH with the electrons reducing molecular oxygen to water. In this way, such pathways could have been utilized, and the full range of functional capabilities ultimately passed down to LECA.

These considerations have implications regarding evolutionary conflict as well. When FECA found itself in a highly favorable environment, oxidative phosphorylation may have been central to its bioenergetics. As suggested by modern aerobic mitochondria, when provided with the necessary materials (e.g., substrate, oxygen, ADP, and phosphate) phosphorylation would be maximal with moderate ROS formation (i.e., "state 3" metabolism of ref. [21]). On the other hand, once all of the ADP is converted into ATP, these mitochondria would continue to oxidize substrate until the trans-membrane proton gradient became maximal and the electron carriers were highly reduced (i.e., "state 4" of ref. [21]). ROS formation would then become maximal. In this context, uncoupler proteins would be an extremely useful adaptation, harmlessly lowering the proton gradient by returning protons to the matrix. Both uncouplers and ADP/ATP carriers (AAC's) belong to a well-defined eukaryotic gene family [24]. Indeed, the latter seem to have evolved from the former, and AAC's still function as proton channels [25]. For protomitochondria in state 4, uncouplers and AAC's have a similar effect, one lowering the proton gradient directly, the other bringing in ADP so that the proton gradient could be diminished by making more ATP. It was of course necessary for FECA's cytoplasm to consume ATP and regenerate ADP and phosphate. In this way, AAC's had a profound impact on the emerging symbiosis.

Nevertheless, uncouplers and AAC's require innovation, which inevitably takes time. Co-opting existing bacterial mechanisms to diminish ROS may have preceded such innovation. In this way, many of the signaling pathways that govern eukaryotic cells may have originated at this time as arbiters of evolutionary conflict [15,26]. Consider for example calcium signaling [27]. In eukaryotic cells, calcium signaling is highly versatile and regulates many different functions [28,29]. By using various pumps, channels, exchangers, and binding proteins, the intracellular concentration of calcium is maintained at a much lower level than that of the extracellular environment. Low intracellular concentrations of calcium are necessary to avoid the precipitation of calcium phosphate. This low intracellular background, however, also allows signaling via pulses or waves of $Ca^{2+}$ originating from the extracellular environment or intracellular stores. Calcium ions are thus a key "second messenger" in multicellular animals. While many aspects of calcium signaling in modern cells may be more recently derived, the process itself may have roots dating back to FECA. Once protomitochondria became endosymbionts, they began to interact directly with the host and only indirectly with the external environment. For instance, mitochondria could congregate near a calcium channel in the cell membrane. In the presence of substrate, a calcium signal could be energized by mitochondria by taking up and expelling the calcium, while at the same time oxidative phosphorylation would increase and ROS would decrease [15,27]. If mitochondria were deprived of substrate, however, the calcium signal would not be energized. In this way, under-resourced mitochondria could signal their metabolic state to the host. Calcium ions, along with ROS and ATP, may have been the tools that mitochondria used to communicate with their hosts [30]. The many facets of calcium signaling in modern cells may thus be vestiges of this ancient host-symbiont interaction.

If the original syntrophic relationship developed in an aerobic environment at least in part, ROS signaling may have arbitrated conflict from the beginning of the evolutionary transition [7]. In a favorable environment with abundant substrate and molecular oxygen, protomitochondria may have phosphorylated at maximal rates, likely outstripping their

metabolic demand. As chemiosmotic products built up in protomitochondria, so too would ROS. High levels of ROS may have triggered unspecified mechanisms that led to the export of these products (Figure 1). In this way, the protomitochondria may have continued to phosphorylate maximally, while ROS were maintained at moderate levels. The concentration of these products in the environment may have been a valuable resource for another microbe, which utilized these products. The benefit to the protomitochondria would be two-fold: first, the concentration of raw materials in the environment would increase, allowing easier uptake, and second, the concentration of products in the environment would decrease, facilitating export. A syntrophic relationship between these microbes would quickly develop. Note, however, that the principal benefit to the protomitochondria was simply getting rid of end products in order to rein in high levels of ROS.

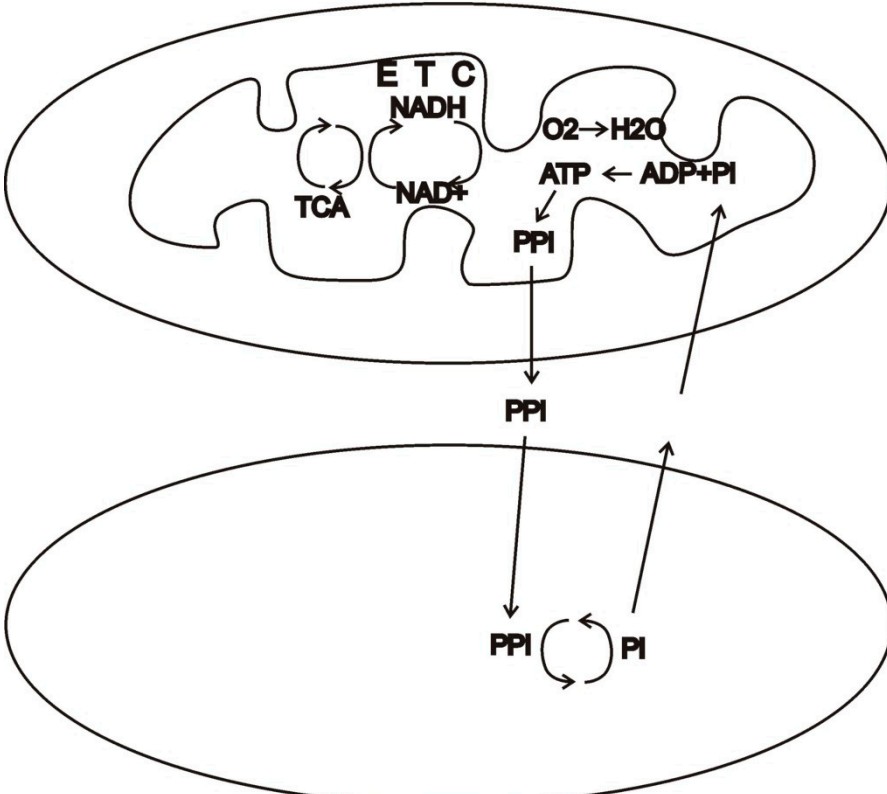

**Figure 1.** Simplified schemata of a possible aerobic interaction between a protomitochondrion and a proto-host. The protomitochondrion (top panel) uses the Krebs or TCA cycle to oxidize substrate and reduce $NAD^+$. NADH is re-oxidized by the electron transport chain (ETC) while oxygen is reduced to water and ATP forms from ADP and $P_i$. Under favorable conditions, the protomitochondria exports $PP_i$ to maintain state 3 metabolism [21]. This export may be triggered by an increase in ROS, which are then maintained at moderate levels [7]. $PP_i$ may be taken up and metabolized by the proto-host, which in turn exports $P_i$.

In the initial syntrophic relationship, whether based on an aerobic chemiosmosis or the anaerobic hydrogen metabolism, or both, cooperation can be maintained by stoichiometry [26]. Neither partner could profit by exporting or importing more or less than their share, since the chemical reactions determine these shares. Once an endosymbiosis was established and ATP became the principal product of mitochondria, however, a mitochondrion could defect from the cooperative relationship by ceasing to export this product, e.g., via loss-of-function mutations. In some environments (e.g., nutrient-scarce ones), this might be selectively neutral or even beneficial. Under conditions of abundant resources, however, defectors that hoard products face the risk of damage or destruction from high

levels of ROS and programmed cell death (Figure 2). ROS signaling pathways may thus have arbitrated evolutionary conflict.

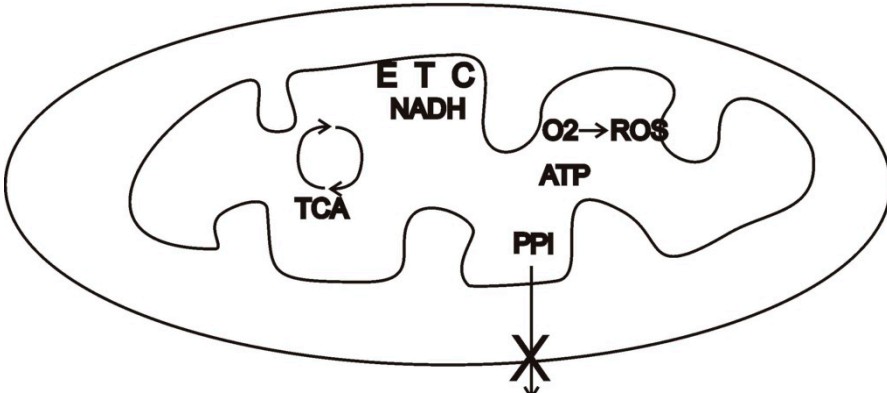

**Figure 2.** Simplified schemata of a protomitochondrion in metabolic state 4 [21]. The protomitochondrion exhibits a loss-of-function mutation and cannot export pyrophosphate (PP$_i$). Under favorable conditions, the protomitochondria continues to oxidize substrate in the Krebs or TCA cycle until all of the ADP is converted to ATP. The electron transport chain (ETC) continues to oxidize NADH until the proton gradient is maximal and the electron carriers of are highly reduced. High levels of reactive oxygen species (ROS) result. In this way, hoarding the products of chemiosmosis may be detrimental to a protomitochondrion, while sharing with a proto-host (Figure 1) may be beneficial.

## 4. Anaerobic FECA in a Euxinic Ocean

Two aspects of hypothetical initial interactions between the partners that comprised FECA are outlined above, one purely anaerobic, the other purely aerobic. In neither case are the sulfur compounds in a euxinic ocean considered. Of course, hydrogen sulfide mimics molecular oxygen and inhibits cytochrome oxidase (complex IV), the terminal electron carrier that reduces molecular oxygen to water. Environments in which molecular oxygen may have been intermittently available (e.g., because of photosynthetic cyanobacteria) would likely have been limited to the narrow photic zone (e.g., [11]). What of FECA that dispersed into the depths? The hydrogen hypothesis is attractive in this context, but some modern mitochondria exhibit other anaerobic pathways.

At least as far as has been determined, modern eukaryotes exhibit a diverse and sophisticated metabolism of sulfur compounds. Hydrogen sulfide can be formed in numerous locations in the cytoplasm and in the mitochondria from sulfur-containing amino acids (cysteine and homocysteine) and other sources [31]. Other reactive sulfur species (RSS) can be generated from various reactions of H$_2$S, for instance, stepwise oxidation to thiyl radical (HS$^\bullet$), hydrogen persulfide (H$_2$S$_2$), persulfide radical (S$_2^{\bullet-}$), and ultimately elemental sulfur (S$_2$) [32]. Parallels to various forms of ROS are apparent, including that many "antioxidant" enzymes act on both RSS and ROS [32]. Modern mitochondria also exhibit a pathway for the breakdown of H$_2$S via its oxidation to sulfate using sulfide:quinone oxidoreductase (SQR), an evolutionarily ancient enzyme [16].

The sulfur metabolism of modern eukaryotes may well be a legacy of the conditions in the Proterozoic oceans under which eukaryotes evolved. For example, Searcy [33] suggests that eukaryotes may have arisen as a syntrophic relationship between sulfur-oxidizing protomitochondria and sulfur-reducing archaea (see also [11]). With a constellation of RSS present in eukaryotic cells [31,32,34], it would seem reasonable to consider their role in scavenging electrons from mitochondrial electron transport. Since a biomedical context largely frames the current focus on RSS, this nevertheless has not been done. In the aerobic cells that are the focus of biomedicine, RSS are usually considered to be electron donors rather than electron acceptors. On the other hand, sulfate-reducing bacteria play a large role in the biosphere-level sulfur cycle [35]. Further, certain RSS, e.g., the thiyl radical and similar molecules, are strong oxidants [34].

In this context, consider malate dismutation, a type of anaerobic mitochondrial metabolism that is widely found in animals [16]. Malate formed in the cytosol from oxaloacetate allows re-oxidizing the NADH from glycolysis. Malate is then imported into mitochondria and a portion of it is oxidized while a portion is reduced. The oxidative branch produces NADH, which must be re-oxidized, and succinate, which may be converted into acetate. The other portion of malate is converted to fumarate, then succinate by running the Krebs cycle backwards. NADH is oxidized by complex I of the electron transport chain. Rhodoquinone carries these electrons to the membrane-bound fumarate reductase which deposits them on fumarate, forming succinate. Succinate may be converted into other products such as acetate and propionate. While there are several other common mechanisms of anaerobic metabolism in eukaryotes that use portions of the electron transport chain, malate dismutation may serve as a useful example (Figure 3).

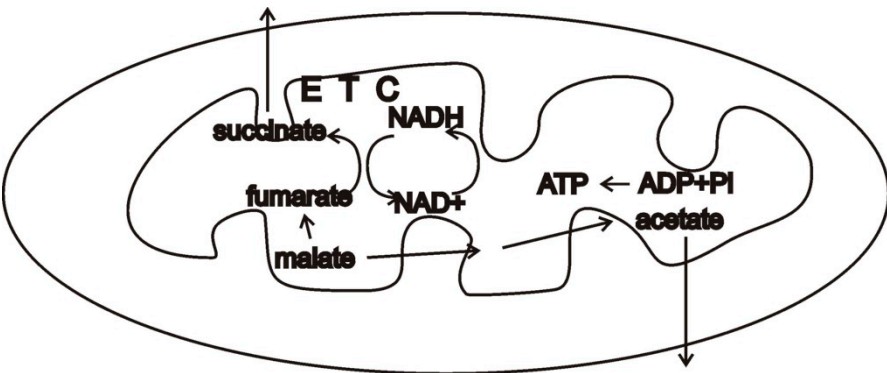

**Figure 3.** Simplified schemata of malate dismutation [16], a possible type of anaerobic metabolism of a protomitochondrion. Malate is oxidized to pyruvate and acetate, reducing $NAD^+$ to NADH. Malate is also reduced to fumarate and succinate, and possibly other products, oxidizing NADH to $NAD^+$ by using a portion of the electron transport chain (ETC). A syntrophic partner that could take up succinate, acetate, and other products would be valuable by diminishing end-product inhibition (Figure 4).

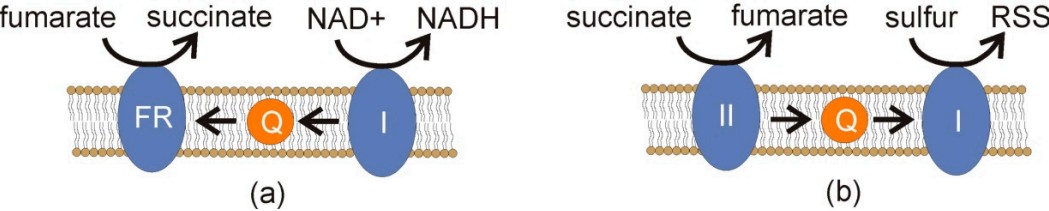

**Figure 4.** Simplified schemata of a scenario for RSS formation under anaerobic conditions. (**a**) As part of the reductive branch of malate dismutation [16], NADH is oxidized to $NAD^+$ by complex I (I) of the electron transport chain (Figure 3). Electrons are carried by rhodoquinone (Q) to fumarate reductase (FR), reducing fumarate to succinate. (**b**) If succinate accumulates, it can inhibit the process in (**a**) and donate electrons to complex II (II) of the electron transport chain, resulting in reverse electron transport in which ubiquinone (Q) carries electrons to complex I (I), and these electrons are ultimately scavenged by sulfur to form RSS. Paralleling Figure 2, such RSS formation would select for exporting succinate and related products to avoid this sort of end-product inhibition.

If end-product inhibition occurs, e.g., the cell has a surfeit of acetate and propionate, a build-up of succinate can occur. Such a build-up can pose risks in the presence of oxygen. When succinate donates electrons to complex II, reverse electron transfer can occur, and electrons flowing to complex I can be cast off to molecular oxygen if it is present, leading to ROS [36]. When oxygen is absent and hydrogen sulfide is present, reverse electron transfer would seem highly likely, since both $H_2S$ and the lack of molecular oxygen will inhibit canonical electron transfer. Under these conditions, RSS might increase to harmful levels

(Figure 4). Thus, there would be strong selection to avoid end-product inhibition in the disposal of succinate, acetate, propionate, and any other products. In this way, a symbiont that took up and metabolized these molecules would be a valuable partner indeed.

These conclusions are unlikely to be unique to malate dismutation. In the euxinic ocean, any pathway for anaerobic metabolism that utilized a portion of the electron transport chain may have similarly produced RSS when subject to end-product inhibition. The scenario for the evolution of calcium signaling outlined above may have similarly occurred under anaerobic conditions with RSS. The effects of chemiosmosis on ROS and RSS may have allowed both aerobic and anaerobic conditions to work together to allow conflict arbitration and mediation and shape the origin of the eukaryotic cell.

## 5. Discussion

Metabolism can lead to sharing, sharing can lead to groups, groups can lead to cooperation, and cooperation can lead to complexity [15]. Such considerations may have figured prominently in minor and major evolutionary transitions, both in the history of life and in more modern settings. This perspective is based on well-established considerations of chemiosmotic metabolism, e.g., the necessity of holding ROS in check. Biomedical research of course is focused on aerobic mitochondria, so these are the best understood. Nevertheless, it is now clear that there are a variety of types of mitochondria—mitosomes, hydrogenosomes, anaerobic mitochondria—with different physiologies, particularly with regard to the use of molecular oxygen and thus the formation of ROS. How do such considerations affect the role of chemiosmosis in mediating or arbitrating evolutionary conflict?

Given the character states of modern mitochondria, facultatively anaerobic mitochondria seem plausible as a common ancestor [16,17]. The picture that emerges of the FECA to LECA transition is thus one of multi-functionality on the part of both the hosts and the symbionts, depending on the environmental conditions. No attempt was made here to build an explicit model of syntrophic interactions. Rather, several well-characterized examples are reviewed in the context of putative multi-functionality. While hydrogen metabolism may have initiated the symbiosis, other metabolic functions—including anaerobic pathways and oxidative phosphorylation as well—must have also occurred. In this way, LECA inherited the full range of eukaryotic metabolic potential, which was subsequently narrowed in most lineages leading to modern eukaryotes [16].

Aerobic mitochondria can suggest that signaling with ROS was central to the origin of mitochondria and, indeed, eukaryotes. Nevertheless, modern anaerobic mitochondria and hydrogenosomes indicate that LECA had other capabilities and may even suggest an anaerobic origin of mitochondria and eukaryotes. The geological history of the earth indicates that molecular oxygen was scarce during the Proterozoic, particularly in the ocean, while hydrogen sulfide was abundant [16]. How can these views, based on the divergent physiology of modern mitochondria, be reconciled?

Signaling with RSS may be a missing piece of the puzzle. Modern cells with aerobic mitochondria nevertheless exhibit a sophisticated sulfur metabolism, although one in which sulfur often functions as an electron donor, while molecular oxygen is commonly an electron acceptor. In the vast, euxinic ocean of the Protozoic, however, RSS may have functioned more like modern ROS, formed by scavenging electrons from electron transport chains when end-product inhibition caused a back-up of electrons. Many of the signaling pathways involving ROS may have first evolved using RSS [31,32]. Much of what aerobic eukaryotic cells now do with ROS may have first been done with RSS. This may include roles in mediating or arbitrating evolutionary conflict [7]. In this sense, ROS and RSS can be viewed as partners in the pathways that alleviated conflict and built the eukaryotic cell.

While much of the above focuses more-or-less exclusively on the evolution of eukaryotes, many of the same ideas may apply more widely to other symbioses, although the conditions of eukaryogenesis are particularly favorable for the actions of chemiosmosis in mediating evolutionary conflict [13,15]. At the same time, symbioses that occurred subsequent to the Proterozoic are likely more clearly impacted by molecular oxygen. In terms

of eukaryotes, as noted above most modern symbioses (e.g., corals, lichens, mycorrhizal fungi) involve oxygenic photosynthesis rather than oxidative phosphorylation. ROS may currently dominate signaling pathways that mediate or arbitrate evolutionary conflict [7], while in the Proterozoic RSS may have had a considerably larger role.

**Funding:** This research received no external funding.

**Acknowledgments:** Helpful comments and suggestions were provided by two reviewers.

**Conflicts of Interest:** The author declares no conflict of interest.

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
