# Peer review of "Reactive Oxygen and Sulfur Species: Partners in Crime"

_oxygen, doi:10.3390/oxygen2040032_

Round 1
Reviewer 1 Report
It is a nice piece and well written. The novel idea is the role of reactive sulfur species (RSS). Nobody linked the potential of SRR production to the origin of mitochondria before, that is my understanding. His main points are that "the signaling pathways involving ROS may have first evolved using RSS" and that "Signaling with RSS may be a missing piece of the puzzle" in the origin of mitochondria. I like the idea, especially because it takes into account the geological history of the Earth in the sense that oxygen might not have been that abundant whereas H2S likely was… The other thing I like is that the scenario takes into account potential multi-functionalities of both partners, depending on the environmental conditions. What I think it is missing is a more explicit comparison to the other hypotheses involving sulfur metabolism (eg, the sulfur-transfer-based model (HS Syntrophy); https://www.nature.com/articles/s41564-020-0710-4); just so the reader appreciates the differences in how sulfur is linked to the origin of mitochondria between previous hypotheses and his hypothesis (his involving sulfur - as RSS, in the arbitration of symbiosis, rather than the symbiosis itself)… Such an explicit discussion would avoid readers having the same concerns in terms of asking about other ecological scenarios… Basically, his point is not so much about the syntrophic relationships per se (in terms of what the partners are exchanging) but rather that if there are metabolic imbalances and end-product inhibition, RSS (instead of ROS; ie, when oxygen is not in the environment) can be produced… and the production of RSS could have been a selective pressure that favoured the maintenance of the association. See "The effects of chemiosmosis on ROS and RSS may have allowed both aerobic and anaerobic conditions to work together to allow conflict arbitration and mediation and shape the origin of the eukaryotic cell" (page 7-8). The other thing I think would be useful to have another figure (based on figure 3) to show the involvement of RSS - similar to Figure 2.
The cross feeding scenario and the removal of end product inhibition by the group is not new to this paper. However, Blackstone's contribution is - started before this paper (in previous paper in Oxygen), is to frame this in terms of the production of ROS (and now RSS) as being a selective pressure that favours the association and prevents the occurrence of cheaters.
Author Response
Thanks to both reviewers for the helpful comments and suggestions. I have carefully considered these comments and revised the manuscript appropriately as discussed below.
“It is a nice piece and well written. The novel idea is the role of reactive sulfur species (RSS). Nobody linked the potential of SRR production to the origin of mitochondria before, that is my understanding. His main points are that "the signaling pathways involving ROS may have first evolved using RSS" and that "Signaling with RSS may be a missing piece of the puzzle" in the origin of mitochondria. I like the idea, especially because it takes into account the geological history of the Earth in the sense that oxygen might not have been that abundant whereas H2S likely was… The other thing I like is that the scenario takes into account potential multi-functionalities of both partners, depending on the environmental conditions.”
Response: Thank you for the kind words.
“What I think it is missing is a more explicit comparison to the other hypotheses involving sulfur metabolism (eg, the sulfur-transfer-based model (HS Syntrophy); https://www.nature.com/articles/s41564-020-0710-4); just so the reader appreciates the differences in how sulfur is linked to the origin of mitochondria between previous hypotheses and his hypothesis (his involving sulfur - as RSS, in the arbitration of symbiosis, rather than the symbiosis itself)… Such an explicit discussion would avoid readers having the same concerns in terms of asking about other ecological scenarios… Basically, his point is not so much about the syntrophic relationships per se (in terms of what the partners are exchanging) but rather that if there are metabolic imbalances and end-product inhibition, RSS (instead of ROS; ie, when oxygen is not in the environment) can be produced… and the production of RSS could have been a selective pressure that favoured the maintenance of the association. See "The effects of chemiosmosis on ROS and RSS may have allowed both aerobic and anaerobic conditions to work together to allow conflict arbitration and mediation and shape the origin of the eukaryotic cell" (page 7-8).”
Response: Yes, this is exactly correct. The goal of the manuscript was not to develop a model of the symbiosis but rather to explore the existing models in terms of the implications for conflict mediation in the absence of molecular oxygen. I have cited the suggested paper in several places as this provides a recent summary of such models. I have also added text to the end of the introduction and the beginning of the discussion to clarify this point.
“The other thing I think would be useful to have another figure (based on figure 3) to show the involvement of RSS - similar to Figure 2.”
Response: While this was discussed in the text, the reviewer is correct that an additional figure would bring this topic into clearer focus. I have added a Figure 4 showing how this might work in the context of malate dismutation, which is elaborated in Figure 3. This figure now reinforces one of the central points of the manuscript, and thus this addition greatly strengthens the manuscript.
Reviewer 2 Report
In this review, Blackstone further expands his ideas on the emergence of complex life forms, cooperation, energetic constraints and reactive species. This topic has already been explored in a very recent review paper by the same author (https://doi.org/10.3390/oxygen2030019). The novelty of the current manuscript is the role of reactive sulfur species. The manuscript is well-written, presenting its key ideas logically and clearly while covering the relevant literature. The review is timely given the increase in the appreciation of the relevance of sulfur species in the evolution of and extant redox signaling pathways. I do have a single minor suggestion, which is to include a figure/scheme illustration the role sulfur species would have played in energetics and redox signalling in the context of cooperation and emergence of complexity.
Author Response
Thanks to both reviewers for the helpful comments and suggestions. I have carefully considered these comments and revised the manuscript appropriately as discussed below.
“In this review, Blackstone further expands his ideas on the emergence of complex life forms, cooperation, energetic constraints and reactive species. This topic has already been explored in a very recent review paper by the same author (https://doi.org/10.3390/oxygen2030019). The novelty of the current manuscript is the role of reactive sulfur species. The manuscript is well-written, presenting its key ideas logically and clearly while covering the relevant literature. The review is timely given the increase in the appreciation of the relevance of sulfur species in the evolution of and extant redox signaling pathways.”
Response: Thank you for the kind words.
“I do have a single minor suggestion, which is to include a figure/scheme illustration the role sulfur species would have played in energetics and redox signalling in the context of cooperation and emergence of complexity.”
Response: While this was discussed in the text, the reviewer is correct that an additional figure would bring this topic into clearer focus. I have added a Figure 4 showing how this might work in the context of malate dismutation, which is elaborated in Figure 3. This figure now reinforces one of the central points of the manuscript, and thus this addition greatly strengthens the manuscript.